# Formation of twelve-fold iodine coordination at high pressure

Yan Liu[1], Rui Wang[2], Zhigang Wang [2], Da Li [1✉] & Tian Cui [1,3✉]

Halogen compounds have been studied widely due to their unique hypercoordinated and hypervalent features. Generally, in halogen compounds, the maximal coordination number of halogens is smaller than eight. Here, based on the particle swarm optimization method and first-principles calculations, we report an exotically icosahedral cage-like hypercoordinated $IN_6$ compound composed of $N_6$ rings and an unusual iodine—nitrogen covalent bond network. To the best of our knowledge, this is the first halogen compound showing twelve-fold coordination of halogen. High pressure and the presence of $N_6$ rings reduce the energy level of the 5d orbitals of iodine, making them part of the valence orbital. Highly symmetrical covalent bonding networks contribute to the formation of twelve-fold iodine hypercoordination. Moreover, our theoretical analysis suggests that a halogen element with a lower atomic number has a weaker propensity for valence expansion in halogen nitrides.

[1] State Key Laboratory of Superhard Materials, College of Physics, Jilin University, Changchun 130012, P.R. China. [2] Institute of Atomic and Molecular Physics, Jilin University, Changchun 130012, P.R. China. [3] School of Physical Science and Technology, Ningbo University, Ningbo 315211, P.R. China. ✉email: dali@jlu.edu.cn; cuitian@nbu.edu.cn

The fundamental chemical properties of a given element strongly depend on its electronic configuration[1]. Generally, to satisfy the octet rule, an element gains or loses electrons to fulfill its outer electron shell, attaining the electronic configuration of the nearest noble gas element[2,3]. An element with a completely filled outer electron shell is highly stable under ambient conditions. However, in a high-pressure environment, the rules of classical chemistry are broken[4,5]. At high pressure, the electronic configuration of an atom will change drastically, enabling matter to exist in a totally counterintuitive chemical regime. For example, the most "simple" metal, Na, undergoes an unusual metal–insulator phase transition at high pressure due to unexpected p–d orbital hybridizations[6]. The noble gas Xe, a chemically inert element with a completely filled shell under ambient conditions, captures electrons under high pressure, becoming negatively charged in Mg–Xe compounds[7]. Group VIIA elements (halogens) with partially filled p shells are usually stable in a −1 charge state at ambient pressure, satisfying the octet rule of chemistry, wherein table salt, NaCl, is the most well-known compound, and its chemistry is well understood. Interestingly, halogens also exhibit hypervalent features in their compounds, such as $XeF_2$ and $CsI_3$, under moderate conditions[8,9]. At high pressure, the classical chemistry rules regarding halogens are further broken. Two fascinating compounds, $Na_3Cl$ and $NaCl_3$, with unusual compositions that violate the octet rule were predicted in theory and confirmed by high-pressure experiments[4]. Furthermore, previous studies have indicated that fluorine, the most electronegative halogen[10], can activate the closed shell of an element in fluoride at high pressure. The completely filled 5p inner shell of Cs can be activated to form unexpected F-rich Cs–F compounds, causing Cs to expand beyond the +1 oxidation state[11]. Hg behaves as a transition metal atom and transfers electrons from its closed-shell 5d orbitals to F atoms in Hg–F compounds at high pressure[12]. Recently, Luo et al.[13] predicted a high-pressure-induced hypervalent interhalogen compound ($IF_8$) with eightfold iodine coordination.

It is well known that among light elements, nitrogen has a high electronegativity[10]. Similar to F atoms, N atoms have the ability to activate the vacant 5d orbitals of iodine under high pressure. However, to date, there have been no reports on the hypervalence of I–N compounds, and only two highly explosive and easily decomposed neutral compounds, $IN_3$ and $NI_3$, have been reported[14,15]. Thus, the study of iodine-nitrogen compounds remains particularly challenging. Although several notable nitrogen-rich nitrides, such as $HeN_4$, $LiN_5$, and $XeN_6$, have been proposed to exist at high pressure[16–19], nitrogen-rich halogen nitrides have not yet been reported. Conventional chemistry rules that apply under ambient conditions are broken under high-pressure conditions; thus, a high-pressure environment is a versatile tool for developing stable materials with unexpected stoichiometries. High pressure endows nitrogen with the ability to interact in a variety of ways, for example, via van der Waals interactions or strong covalent bonding with other atoms[16,20–22]. Therefore, potential high-pressure, nitrogen-rich iodine nitrides with exotic properties are worthy of exploration.

In this study, taking iodine as an example, we investigated the feasibility of using high pressure to stabilize nitrogen-rich halogen-nitrogen compounds and further explored variations in the pressure-induced energy of the outer shell of halogens. We utilized crystal structure analysis via particle swarm optimization (CALYPSO) together with first-principles simulations to perform an extensive search on the structures of the selected stoichiometries of $I_xN_y$ ($y/x + y = 0 - 1$) from 0 to 150 GPa. An unexpected icosahedral, cage-like, hypercoordinated, and hypervalent nitrogen-rich compound, $IN_6$, with a hexagonal $R\bar{3}m$ space group is predicted to be stable above 100 GPa. The iodine atom in $IN_6$

has unusual 12-fold coordination, which is the largest possible coordination state of halogen atoms and is reported for the first time. The combination of high pressure and strong $N_6$-ring ligands reduces the energy level of the iodine 5d orbitals, forcing them to become part of the valence orbital and resulting in the formation of covalent bonds between the iodine and nitrogen atoms. Furthermore, the formation mechanism of this novel icosahedral, cage-like, hypercoordinated, and hypervalent $IN_6$ compound was investigated.

## Results

**Stability of I–N compounds.** Structure prediction simulations for iodine-nitrogen compounds with 1–4 formula units were first performed using the PSO method through the CALYPSO code for each composition at various pressures in the range of 0–150 GPa. These searches were performed without any experimental information. Then, Eq. (1) was employed to calculate the average atom formation enthalpy for each composition under various pressures. The corresponding stable phases of solid nitrogen and iodine at different pressures were chosen as reference structures, where nitrogen adopted $Pa\bar{3}$, $P4_12_12$, and $I2_13$ structures[23,24], and bulk iodine adopted $Immm$, $I4/mmm$, and $Fm\bar{3}m$ structures[25–27].

$$\Delta H_f(I_xN_y) = [H(I_xN_y) - xH(I) - yH(N)]/(x+y) \qquad (1)$$

Stable I–N compounds were determined by using convex hull construction, as shown in Fig. 1. The previously reported halogen nitride $I_3N$ was thermodynamically metastable in our structure search. At elevated pressures, the predicted $IN_3$ and $IN_6$ compounds among the candidate compositions matched the convex hull and were deemed stable and synthesizable. $IN_3$ emerges at 80 GPa and stabilizes in a monoclinic structure (space group $C2/m$) containing four units. The nitrogen atoms from the diatomic molecule $N_2$, which occupies two distinct sites that are parallel and perpendicular to the b-axis of the monoclinic unit cell. The N−N distances are equal to 1.26 and 1.29 Å, respectively, and are slightly larger than the length of the double bond (1.25 Å) in $N_2H_2$. Electronic band structure calculations revealed the metallic nature of $C2/m$-$IN_3$. Detailed information on the predicted structures is presented in Table 1, Fig. 1, and

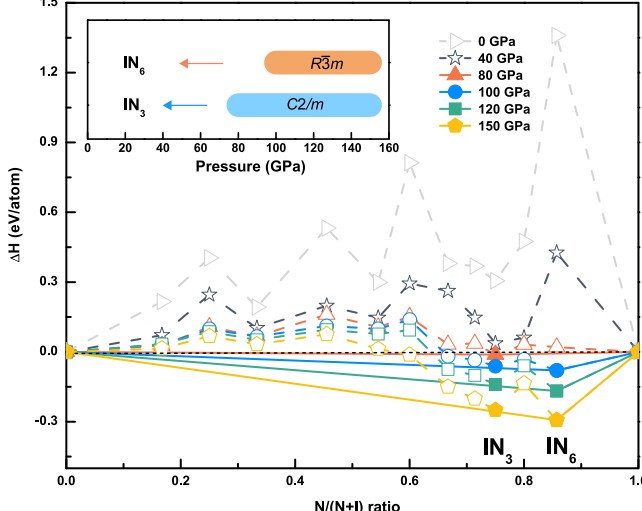

**Fig. 1 Phase stabilities of I–N compounds.** The formation enthalpy of various I–N compounds under various pressures. The dotted lines connect the data points, and the solid lines denote the convex hull. The stable pressure ranges for $IN_3$ and $IN_6$ are shown in the inset. Source data are provided as a Source Data file.

Fig. 2 of the Supplementary Information. Subsequently, our analyses focus on $IN_6$ only because it exhibits nitrogen-rich, icosahedral, cage-like features and unexpected 12-fold coordination.

**Unexpected icosahedral $IN_6$.** The global search predicts that $IN_6$ stabilizes in a hexagonal structure (space group $R\bar{3}m$) at 100 GPa, referred to as $R\bar{3}m$-$IN_6$, and acts as an archetype structure in which pressure promotes the formation of nitrogen-rich halogen-nitrogen compounds. The phonon dispersion calculations suggest the absence of imaginary modes throughout the entire Brillouin zone, indicating that $R\bar{3}m$-$IN_6$ is dynamically stable at the considered pressures (Supplementary Fig. 3). Within $R\bar{3}m$-$IN_6$, N and I atoms occupy Wyckoff 18 h and 3a sites, respectively. Nitrogen atoms exist in the form of polymeric $N_6$ rings, which has been reported in previous nitrogen-rich compounds[18,19,28,29]. Here, it adopts an armchair-like, puckered $N_6$ ring structure similar to the typical $S_6$ ring of sulfur (Fig. 2a)[30]. All six N–N bonds in the $N_6$ units of $R\bar{3}m$-$IN_6$ are identical because all the N–N bond lengths are equal to 1.36 Å at 100 GPa. This bond length is much closer to the single bond lengths (1.35 Å) observed in cg-N at 115 GPa[23], indicating the presence of N–N single bonds. The puckered $N_6$ rings form linear arrays embedded in an iodine lattice, exhibiting an intriguing symmetrical I–$N_6$–I sandwich structure, as shown in Supplementary Fig. 4. In $R\bar{3}m$-$IN_6$, each N atom of an armchair-like $N_6$ ring is coordinated by two neighboring I atoms (Supplementary Fig. 5). The I atom forms 12-fold coordination with 12 neighboring nitrogen atoms from 8 armchair-like $N_6$ rings (Fig. 2b), including 6 nearest neighboring N1 atoms in the ab plane and 6 second-nearest neighboring N2 atoms along the $c$ direction, constituting an $IN_{12}$ icosahedron with $D_{3d}$ point group symmetry (Fig. 2c, Supplementary Table 2). The I–N distances in the ab plane and along

the c direction in the $IN_{12}$ icosahedron are equal to 2.23 and 2.27 Å, respectively, and are similar to the summation of the covalent radii of I and N atoms (2.15 Å)[31], indicating that covalent interactions occur between the I and N atoms. The covalent properties of I–N bonds can also be revealed by a more intuitive approach, the electron localization function (ELF), which is a measure of relative electron localization[32,33], and it maps values in the range from 0 to 1, where 0.5 represents the situation in a homogeneous electron gas. Large ELF values usually occur in regions with a high tendency of forming electron pairs, corresponding to bonds, lone pairs of electrons, and electron shells[34,35]. As shown in Supplementary Fig. 6, the maximum ELF value of ~0.9 between the iodine and nitrogen atoms indicates the existence of covalent bonds. In addition, the shortest I–I distance (3.73 Å) is more than double the covalent radius of I (1.4 Å)[31], excluding any possibility that I–I bonds form.

**Chemical bonding and electronic properties of $IN_6$.** Next, topological analysis of the all-electron charge density of $R\bar{3}m$-$IN_6$ was performed by using atoms-in-molecules theory to quantitatively describe the chemical bonding behavior[36]. The charge density distribution and its principal curvatures (the three eigenvalues of the Hessian matrix) at the bond critical points reveal information about the bonding types and properties (Supplementary Fig. 7). The sign of the Laplacian of the electron density ($\nabla^2\rho(r)$) indicates whether the density is locally concentrated (negative) or depleted (positive)[37]. Previous works have shown that $\nabla^2\rho(r)$ values at critical points can efficiently reflect the strength of covalent bonds[16,18,38,39]. As expected, all the values of these I–N bonds in $R\bar{3}m$-$IN_6$ are negative (Table 1), indicating obvious covalent interactions between I and N atoms, but the $\nabla^2\rho(r)$ of the I–N1 (−0.48) and I–N2 (−0.41) bonds suggest two different bond strengths and indicate that the covalent I–N2 bond is slightly weaker than the covalent I–N1 bond. These results are in excellent agreement with those derived from bond length and ELF analysis.

Electronic structures are critical to understanding the nature of the formation mechanism of a material. As illustrated in Fig. 3a, the nonzero projected density of states (PDOS) at the Fermi level indicates the metallic features of $R\bar{3}m$-$IN_6$ (also see the band structure of $R\bar{3}m$-$IN_6$ in Supplementary Fig. 8). Both the I and N atoms contribute to the metallic properties of $R\bar{3}m$-$IN_6$. The PDOS of the 2s and 2p orbitals of N overlap in the whole energy range, reflecting the strong orbital hybridization between them. In

**Table 1 Bond critical point data for the I–N bonds in $R\bar{3}m$-$IN_6$ at 100 GPa.**

| Bond type | Length (Å) | $\nabla^2\rho(r)$ |
|---|---|---|
| I–N1 | 2.23 | −0.48 |
| I–N2 | 2.27 | −0.41 |

The negative Laplacian values ($\nabla^2\rho(r)$) indicate that covalent bonds are formed. Source data are provided as a Source Data file.

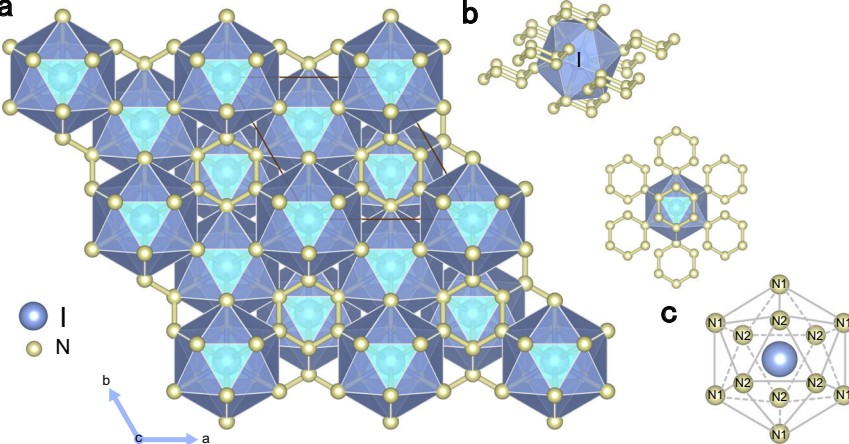

**Fig. 2 Schematic representation of $IN_6$ crystal structure. a** $IN_6$ in an $R\bar{3}m$ structure at 100 GPa. **b** The 12-fold coordination of iodine ($IN_{12}$) coordinated with nitrogen atoms from 8 armchair-like $N_6$ rings. **c** Schematic representation of the $IN_{12}$ icosahedron structure with $D_{3d}$ point group symmetry. N1 and N2 represent the nitrogen atoms in the short and long I–N bonds, respectively.

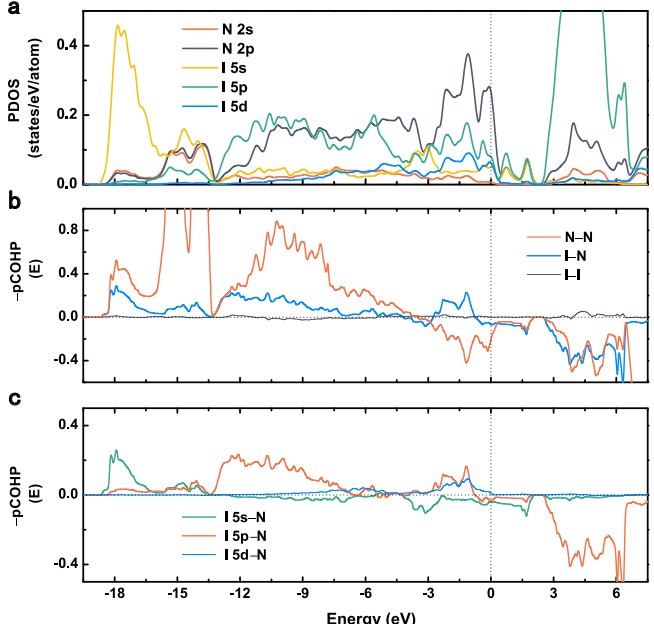

**Fig. 3 Electronic structure of $R\bar{3}m$-$IN_6$ at 100 GPa. a** Projected density of states (PDOS). The Fermi energy is set to zero. **b** Crystal orbital Hamilton population (COHP) of N–N, I–N, and I–I pairs. **c** The projected COHP (pCOHP) of I–N pairs. Positive and negative –pCOHP values denote bonding and antibonding interactions, respectively. Source data are provided as a Source Data file.

hypothetical planar hexazine $N_6$, each N atom coordinates with two neighboring nitrogen atoms, forming two σ bonds via $sp^2$ orbitals and leaving one lone pair of electrons[28,40,41]. However, in $R\bar{3}m$-$IN_6$, the nitrogen atom coordinated with two N and two I atoms forms fourfold coordination and is denoted $AX_4$ in valance shell electron pair repulsion (VSEPR) theory[42], where A and X represent the central atom (N) and its neighboring coordinated atoms, respectively (Supplementary Fig. 9). According to VSEPR theory, the directed valency could rationalize the local structural environments. Combined with the abovementioned single-bond feature of the N–N bond, the nitrogen atom in armchair-like $N_6$ is $sp^3$-hybridized rather than $sp^2$-hybridized, different from that in the hypothetical planar hexazine $N_6$ ring. In hypercoordinated $IN_6$, each N atom has two remaining $sp^3$ orbitals interacting with the I atom. Therefore, each I atom is coordinated with 12 N atoms, where each N $sp^3$ orbital points toward the center I atom, forming the $IN_{12}$ icosahedron.

Unlike N 2p, I 5s, and I 5p, I 5d contributes abnormally to the density of states at the Fermi level. The I 5d band undergoes significant dispersion near the Fermi level and mixes with the 5s/p orbitals of iodine and 2p orbitals of nitrogen (Fig. 3a). Therefore, I 5d orbitals that are originally vacant in a single iodine atom become valence electron orbitals and participate in bonding between nitrogen and iodine atoms in $R\bar{3}m$-$IN_6$. The crystal orbital Hamilton population (COHP), an energy-resolved partitioning scheme of the band structure energy on the basis of atomic and bonding contributions, provides convincing evidence for the above analyses. Negative values of COHP indicate bonding, while positive values indicate antibonding behavior[43,44]. Considering all valence orbitals, the bond strength between two interacting atoms can be visualized by investigating the complete COHP between them, and the integrated COHP (ICOHP) is used as a qualitative measure of mainly covalent bond strength, where the greater the negative value is, the stronger the

covalent interaction. As shown in Fig. 3b, the negative COHP observed below the Fermi level means that N–N and I–N interactions are responsible for the structural stability. The resulting ICOHPs of N–N and I–N pairs up to the Fermi level are −12.75 and −2.04 eV, respectively, which further confirms that strong covalent bonding interactions occur between I and N atoms inferred by the ELF and topological analysis. In general, the 5s electrons in iodine are regarded as the inner valence electrons, and iodine makes extensive use of its 5p electrons during bonding. The energy of the 5d orbitals in iodine is higher than that of the 5s orbitals (17.9 eV) and 5p orbitals (8.2 eV)[45]. Consequently, the 5d orbitals are unoccupied orbitals and are not considered an important factor contributing to the chemical behavior of iodine under ambient conditions[46]. However, high pressures can effectively modulate the overlap of various orbitals, particularly the 5d orbitals of iodine, because 5d orbitals have more radial nodes than 5s and 5p orbitals[47]. Therefore, the energy difference between the 5s (5p) and 5d orbitals of a single I atom can be reduced under high pressure. Additionally, the PDOS of $IN_0$ after hypothetically removing its N atoms indicates that the existence of a highly electronegative substituent field created by nitrogen in $R\bar{3}m$-$IN_6$ can further reduce the energy differences among the 5s, 5p, and 5d orbitals of iodine (Supplementary Fig. 10). This is because the $N_6$ rings attract electrons from iodine, which results in the central iodine having a positive effective nuclear charge. This ensures that iodine has a strong capacity to attract its outer orbital electrons.

To gain insight into the formation mechanism of hypercoordinated $IN_6$ under high pressure, we investigated the composition of molecular orbitals (MOs) for I 5s, 5p, and 5d orbitals interacting with N 2s and 2p orbitals in an icosahedral cage-like $IN_{12}$ fragment. We cut the $IN_{12}$ molecular fragment from the crystal structure of $R\bar{3}m$-$IN_6$. The symmetry analysis indicates that the $IN_{12}$ molecular fragment consisted of six N1 and six N2 atoms, has a $D_{3d}$ point group symmetry (Fig. 2c). Therefore, according to the $D_{3d}$ point group symmetry, the MOs of the icosahedral cage-like $IN_{12}$ fragment can be described by the orbital interaction of I and $N_6$ ring (see Supplementary Fig. 5). The valence electrons of iodine and nitrogen participate in the formation of chemical bonds in $R\bar{3}m$-$IN_6$. Two valence electrons of nitrogen form two N–N single bonds $\sigma(sp^3, sp^3)$ with two neighboring nitrogen atoms in the $N_6$ ring. The remaining three electrons of the nitrogen atom in two $sp^3$ orbitals are used to interact with the iodine atom. Therefore, a total of 25 valence electrons occupies the MOs of the icosahedral cage-like $IN_{12}$ fragment in the $R\bar{3}m$-$IN_6$ according to the rules of the Aufbau principle (filling from lowest to highest energy), Hund's rules (maximum spin multiplicity consistent with the lowest net energy), and the Pauli exclusion principle (no two electrons with identical quantum numbers). The component analysis of electron-occupied MOs is employed by the Amsterdam density functional (ADF) package[48], and MO energy level diagram is shown in Fig. 4 (the detail also see Supplementary Table 3). It is noteworthy that the designations of the bonding, nonbonding, and antibonding nature are based on the symmetry of orbitals, which will not change under high pressure. The MOs energy level diagram of the icosahedral cage-like $IN_{12}$ fragment reveals that the valence electrons of I and N in $N_6$ ring have strong participation in the MOs of the $IN_{12}$ fragment containing 25 electrons. Note that nine MOs of $IN_{12}$ fragment at the high-lying energy level mainly originate from the combination of the $E_g$ and $A_{1g}$ orbitals of I 5d and $N_6$ ring components. Similarly, the $E_u$ and $A_{2u}$ orbitals of I 5p also participate in the orbital interaction with the orbitals of the $N_6$ ring component. The $A_{1g}$ orbitals of I 5s and $N_6$ ring components form bonding $A_{1g}$ and antibonding $A_{1g}^*$

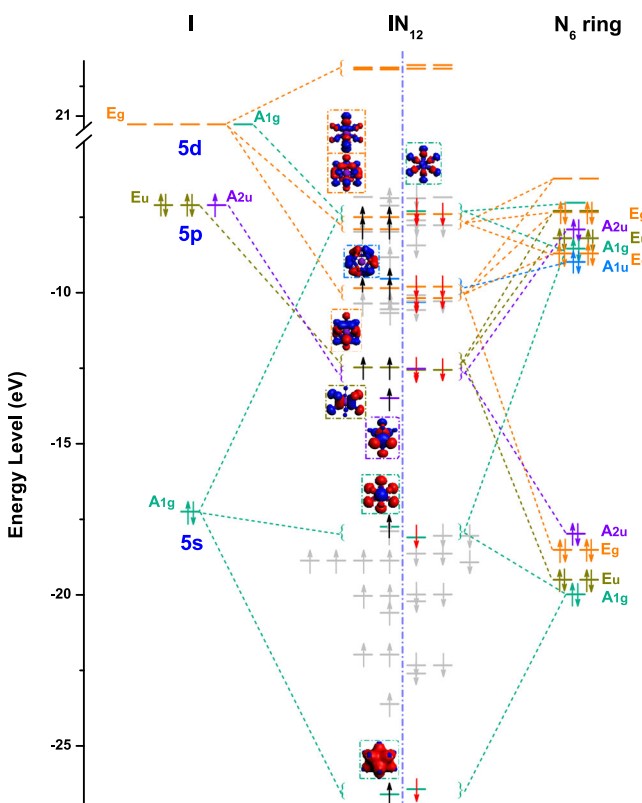

**Fig. 4 Molecular orbital (MO) energy level diagram.** The icosahedral cage-like $IN_{12}$ molecular fragment with $D_{3d}$ point group symmetry. Besides gray annotated MOs of icosahedral cage-like $IN_{12}$, others mainly consist of I and $N_6$ ring components.

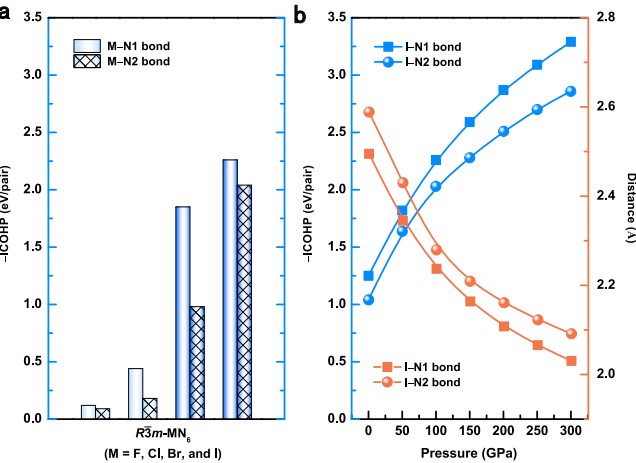

**Fig. 5 Bonding analysis of halogen nitrides. a** Negative integrated crystal orbital Hamilton population (–ICOHP) of the M–N pairs in $R\bar{3}m$-$MN_6$ (M = F, Cl, Br, and I) compounds at 100 GPa. The optimized atomic geometries are shown in the insets. **b** Pressure-induced variations in –ICOHP and bond lengths for the two types of I–N bonds in $R\bar{3}m$-$IN_6$. A positive –ICOHP indicates the presence of bonding states. Source data are provided as a Source Data file.

orbitals of $IN_{12}$ fragment at the low-lying energy levels. These results confirm that I 5d orbitals become a part of the valence orbitals and participate in the bonding interactions. The unique highly symmetrical icosahedral $N_{12}$ cage and the participation of I 5d orbitals in the bonding interactions contribute to the formation of 12-fold iodine hypercoordination.

**Formation mechanism of halogen nitrides.** To gain a comprehensive understanding of halogen nitrides, the possible existence of other high-pressure halogen nitrides was investigated. A model, hypothetical $R\bar{3}m$-$MN_6$ system in which all I atoms were replaced with F, Cl, or Br atoms (astatine is not discussed due to its radioactivity) was constructed. An analysis of the ELF of $MN_6$ revealed that obvious M–N covalent interactions occur in $BrN_6$ and $IN_6$ (Supplementary Fig. 11). This is attributed to the participation of d orbitals of Br and I, which can be confirmed by electronic structure calculations (Supplementary Figure 12). The d orbitals (4d for Br and 5d for I) contribute to the DOS below the Fermi level. Nitrogen, as the coordinate atom, can affect the effective nuclear charge of the central atom, thereby changing the energy of the valence orbitals of the central atom[49]. The F atom has the smallest atomic number in halogen and does not contain d orbitals in its valance shell. The electronegativity of Cl is higher than that of N (Supplementary Table 4)[10], and its effective nuclear charge is reduced in the interaction with N, excluding the possibility of 3d orbitals participating in bonding. Therefore, the N atoms in $FN_6$ and $ClN_6$ form polymeric nitrogen structures instead of connecting with halogen atoms (Supplementary Fig. 13). Each N atom forms three σ bonds with neighboring N atoms via $sp^3$ orbitals, leaving one lone pair of electrons ($AX_3E_1$ in Supplementary Fig. 9, E represents lone pair electrons), which

is responsible for the stability of the system. As the atomic number increases, the electronegativity of the halogen gradually decreases and is even lower than that of nitrogen (Supplementary Table 4)[10], making the effective nuclear charge of the halogen atom increase in the interaction with nitrogen[49]. The outer electrons are drawn towards the nucleus, and the outer orbital energies become more negative. Consequently, the d orbitals of Br and I atoms in $BrN_6$ and $IN_6$ participate in bonding (Supplementary Fig. 12). The propensity of covalent interactions between nitrogen and halogen atoms can be evaluated by calculating their average −ICOHP values. It increases with increasing halogen atomic number (Fig. 5a and Supplementary Table 5). Generally, the principal quantum number n determines the size and energy of the orbital for a given nucleus; as n increases, the size of the orbital increases[50]. When comparing different halogen atoms, the 5d orbitals of I (large principal quantum number) are more extended than the 4d orbitals of Br (small principal quantum number), resulting in a stronger interaction with the icosahedral cage composed of N atoms. The higher principal quantum number makes valence orbitals of halogen farther from the nucleus, and the energy difference among them is smaller. Therefore, the I atom is more likely to form covalent bonds due to the combined effects of high pressure and strong ligands ($N_6$ rings). Notably, the covalent interactions between I and N atoms become stronger with increasing pressure because the −ICOHP values between I and N atoms increase and the bond lengths of I–N bonds decrease with increasing pressure, as shown in Fig. 5b.

Using DFT calculations in conjunction with the CALYPSO method, a global search of halogen nitrides is performed to investigate the nitrogen-induced hypercoordinated feature of halogen under high pressures. An exotically icosahedral cage-like hypercoordinated $IN_6$ compound composed of unusual iodine–nitrogen covalent bonds is predicted at high pressure for the first time. The coordination number of iodine in $IN_6$ is 12, which is the maximal value among the reported halogen compounds, much larger than that of the known neutral iodine fluoride $IF_7$ (7) and theoretically predicted high-pressure $IF_8$ (8). The high pressure and the presence of $N_6$ rings reduce the energy level of the 5d orbital of iodine, making them part of the valence orbital. Highly symmetrical covalent bonding networks

contribute to the formation of twelve-fold iodine hypercoordination. Moreover, a halogen element with a lower atomic number has a weaker propensity for valence expansion in halogen nitrides. Our work provides deep insight into the understanding of halogen chemistry under high pressure.

## Methods

To design reasonable structures, PSO, as implemented in the CALYPSO code[51,52], was performed. PSO has been successfully applied in numerous predictions regarding novel compounds and structures over the past decade[53,54]. In the PSO simulations, underlying structure relaxations were performed within the framework of density functional theory (DFT)[55] using the Perdew–Burke–Ernzerhof (PBE) generalized gradient approximation (GGA)[56] implemented in the Vienna ab initio simulation package[57]. Projector augmented wave[58] pseudopotentials with $5s^2 5p^5$ and $2s^2 2p^3$ valence configurations were chosen for I and N atoms, respectively. Reciprocal space was sampled by a fine grid of $5 \times 5 \times 6$ k-points in the Brillouin zone[59], and the tested cut-off energy for the plane-wave expansion of the wave function was set to 520 eV. Phonon calculations were performed by using the direct supercell method implemented in the PHONOPY program[60]. Chemical bonding analyses were carried out with the COHP method, as implemented in the LOBSTER package[43]. All geometries were optimized using the conjugate gradient method until an energy convergence of $10^{-6}$ eV was satisfied, and none of the residual Hellmann–Feynman forces exceeded $10^{-3}$ eV/Å. The component analysis of electron-occupied MOs of $IN_{12}$ was performed using the ADF package[61]. The analysis was implemented at GGA using PBE functional[56]. Crystal structures and the ELF were drawn using VESTA software[62].

## Data availability

The authors declare that the main data supporting our findings of this study are contained within the paper and Supplementary Information. Source data are provided with this paper. All other relevant data are available from the corresponding author upon request. Source data are provided with this paper.

## Code availability

CALYPSO code is free for academic use, by registering at http://www.calypso.cn. The other scripts are available from the authors upon request.

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

## Acknowledgements

D.L. acknowledges funding from the National Natural Science Foundation of China under Grant nos. 12174141 and 91745203. T.C. acknowledges funding from the National Key R&D Program of China under Grant no. 2018YFA0703400. The calculations were supported by the High-Performance Computing Center of Jilin University, China.

## Author contributions

D.L. designed the project. Y.L., R.W., and D.L. performed the calculations. D.L., Y.L., R.W., Z.W., and T.C. analyzed the data. D.L., Y.L., Z.W., and T.C. wrote the paper.

## Competing interests

The authors declare no competing interests.
