## [Peer Review File · Nature Communications]

Editorial Note: Parts of this peer review file have been redacted as indicated to maintain the confidentiality of confidential information.

Reviewer #1 (Remarks to the Author):

The manuscript presents an interesting and original study of a new hypercoordinated compound IN6 at high pressure. The new compound has an unusual structure with twelve-fold coordination of iodine atoms and due to the presence of nitrogen rings it has a remarkably high energy density. The authors perform a very detailed analysis of bonding including comparison with analogous compounds where iodine is replaced by another halogen atom. The analysis is convincing and the paper is well and clearly written. In my opinion the results are of interest for the high-pressure community while at the same time they represent a valuable contribution to our understanding of the chemistry of halogens. For the above reasons I recommend the manuscript for publication in Nature Communications.

Reviewer #2 (Remarks to the Author):

This manuscript details particle swarm search calculations of novel I-N solid phases under high pressure. In particular, a new IN6 phase is identified and studied in depth. Overall, I found the paper well constructed and represents very interesting results. I have a few comments below that I believe will strengthen the paper should the authors choose to follow them.

-The results for the enthalpy of formation appear to use the enthalpy of elemental I and N as a reference. If so, this could be a serious mistake in that the ambient states of these materials are solid iodine and gas phase N₂. I strongly recommend recomputing the convex hull with this in mind.

-There is use of a lot of acronyms throughout the paper, which makes the analysis extremely confusing and hard to follow. Techniques like ELF and COHP are not standard in all fields, and their physical meaning is not always apparent. Consequently, a lot of the important chemical conclusions in the paper regarding hybrid orbitals and their contributions to bonding states, iodine acting like a transition metal, etc., are nearly impossible for the reader to verify. This is particularly important for a journal like Nat Comm, which has a very broad and general readership. This poses a big complication, and makes it seem like this manuscript might fare better in a more specialized journal. To be clear, I would recommend a more thorough discussion of the uses of ELF and COHP, and more clear discussion about how the computed results support the authors' claims. I also strongly recommend redoing Figs. 3 and 4 so they are easier to read and the reader can more easily see how these results support the states conclusions.

Reviewer #3 (Remarks to the Author):

This manuscript presents DFT-predictions of iodine-nitrogen compounds stabilized under high pressure conditions. The structure mostly focused on can be formally thought of as $[N_6]^{2-}$ rings that are coordinated by $[I]^{2+}$ cations. I have reviewed this work once before, for [redacted] and the authors may recognize some of my comments.

Whereas the predicted chemical structure is quite interesting, the analysis of the electronic structure underlying it is subpar. The authors attempt to throw everything at the poor compound: MO theory, ELF, VSEPR, ideas of octet rules and electronegativity, radii, analyses of the DOS, orbital hybridization, and QTAIM. Unfortunately, a rather shallow understanding of all of the mentioned theories and methods are demonstrated. It would strengthen the work if the authors better learned and focused on a few perspectives of analyzing electronic structure rather than be all over the map. The text is full of statements and loose conclusions that are not supported by calculations or proper explanations. As a consequence, I cannot recommend publication of this work.

It is claimed that iodine behaves as a transition metal under pressure. Yes, it would appear that I 5d levels may become occupied. Based on previous work on the topic of compressed heavy main group elements, the latter is expected. However, the bonding of I is still predominately governed by p-states, which is not the case for any transition metal. The DOS is (as expected) dominated by the 2p levels of nitrogen. And whereas the 5d come down with pressure, and the interaction with N 2p levels is now quantified to some degree, why do these interactions give rise to 12 fold-coordination? Is the 5d really essential for this?

It is in this work (still) claimed that neutral N_6 is planar, but that is not true. There is extensive literature on the this topic (N_6 chemistry). Higher levels of theory show that the planar geometry is unstable. Why do the authors think the neutral molecule is planar? Could it be they simply compute it in that geometry at a low level of theory, and don't check the alternatives, or read the literature?

The authors classify the N-I bonding as:

“a coordinate covalent (dative bond) is formed, which is a kind of two-center-two electron covalent bond in which the two bonding electrons are contributed by a single atom”. Granted this used be even more confused in the last version of the manuscript, I still do not follow. On page 5 it is claimed that three of the 5d orbitals of I are vacant and can serve as electron acceptors for the lone pair electrons on the N_6 rings. At the same time, the authors show how the I atom is oxidized (but that is mentioned elsewhere). And why only three orbitals? All the 5d are formally unoccupied in the

isolated atom, but in the extended system they all form a band, and some degree of all of them become occupied in the interaction with N. Iodine is further claimed to exhibit sp^3d^2 hybridization, but this is not substantiated in any way. The discussion regarding the “octahedral split” is confusing at best. Which are the directions indicated, and how do they relate to the coordination geometry?

Figure 3 shows some sort of orbital splitting diagram (where phases have been omitted). But this diagram is not connected in any relevant way to the analysis, nor is it apparent how it could be. The SI shows an MO diagram of the pi-system of the (unstable) planar N₆, and how the addition of 2e⁻ and the puckering, confusingly enough, does absolutely nothing to the MOs. I recommend the authors instead provide a figure of the puckered N₆ and attempt to explain the effect of occupation by 2 extra electrons. If the authors are able, it would strengthen the work if the interactions of these levels with the I 5p and 5d shell could be shown explicitly in a model compound. If they are not able, which I suspect, then I suggest they instead forgo trying to discuss the bonding in terms of interacting orbitals. Maybe instead more matter of fact list atomic charges and focus on describing the geometry. The COHP looks in order, but the conclusions drawn from it are not, or are taken too far. Simply stating that there is 5d orbital occupation may be enough, and that there is bonding interactions between N and I. The work would be cleaner, and less filled with unsubstantiated claims, if it simply communicated: “we predict that this compound, special in many ways, should be stable at high pressure. Here are some basic bonding properties”. That would be interesting to know for experimentalists. The work could then be shortened by half. One way or another, to be acceptable for publication, this work needs to be significantly toned down and clarified.

The literature is lacking overall. To name just a few examples: Where does the covalent radii come from? Is there any work that has validated the use of the Laplacian of the electron density as an indicator for covalent bonds under high pressure? One could at minimum expect a citation to work showing this under ambient conditions. The lacking N₆ references.

It is claimed that the described compound is a potential energetic material. What happens to the discussed structure when it is relaxed to ambient conditions? Is it still kinetically stable? If not, all such claims should be removed. And if it is stable at ambient conditions, how is the energy density computed? If you compare energy densities of a compressed material with conventional explosives at ambient conditions, it is hardly a fair comparison.

What are the I and N reference phases used for the construction of the convex hull? I hope they are not the same throughout, since the ground state of both elements change with compression. This needs to be specified.

I have several other questions or objections that I will leave out. The bottom line is that the predicted compound may be interesting, but the analysis of it is very far from clear. Either way, I do not recommend this work be further considered by Nature Communications.

Responses to reviewer #1

Comments to the Author

The manuscript presents an interesting and original study of a new hypercoordinated compound IN_6 at high pressure. The new compound has an unusual structure with twelve-fold coordination of iodine atoms and due to the presence of nitrogen rings it has a remarkably high energy density. The authors perform a very detailed analysis of bonding including comparison with analogous compounds where iodine is replaced by another halogen atom. The analysis is convincing and the paper is well and clearly written. In my opinion the results are of interest for the high-pressure community while at the same time they represent a valuable contribution to our understanding of the chemistry of halogens. For the above reasons I recommend the manuscript for publication in Nature Communications.

Author reply:

Thank you. We are very pleased with your recognition of our work.

Responses to reviewer #2

Comments to the Author

This manuscript details particle swarm search calculations of novel I-N solid phases under high pressure. In particular, a new IN_6 phase is identified and studied in depth. Overall, I found the paper well constructed and represents very interesting results. I have a few comments below that I believe will strengthen the paper should the authors choose to follow them.

1. The results for the enthalpy of formation appear to use the enthalpy of elemental I and N as a reference. If so, this could be a serious mistake in that the ambient states of these materials are solid iodine and gas phase N_2 . I strongly recommend recomputing the convex hull with this in mind.

Author reply:

Thank you very much for your suggestions. We have carefully examined the calculation of the enthalpy of formation and found that we chose the corresponding stable phases of solid nitrogen and iodine at different pressures as reference structures, where nitrogen adopted $Pa\bar{3}$, $P4_12_12$ and $I2_13$ structures,^{1,2} and bulk iodine adopted $Immm$, $I4/mmm$ and $Fm\bar{3}m$ structures.³⁻⁵ The description “elemental I and N as a reference” in the original manuscript is wrong, and we are very sorry for this negligence.

Fig. 1. Formation enthalpy of various I-N compounds under various pressures. The dotted lines connect the data points, and the solid lines denote the convex hull. The stable pressure ranges for IN_3 and IN_6 are shown in the inset.

2. There is use of a lot of acronyms throughout the paper, which makes the analysis extremely confusing and hard to follow. Techniques like ELF and COHP are not standard in all fields, and their physical meaning is not always apparent. Consequently, a lot of the important chemical conclusions in the paper regarding hybrid orbitals and their contributions to bonding states, iodine acting like a transition metal, etc., are nearly impossible for the reader to verify. This is particularly important for a journal like Nat Comm, which has a very broad and general readership. This poses a big complication, and makes it seem like this manuscript might fare better in a more specialized journal. To be clear, I would recommend a more thorough discussion of the uses of ELF and COHP, and more clear discussion about how the computed results support the authors' claims. I also strongly recommend redoing Figs. 3 and 4 so they are easier to read and the reader can more easily see how these results support the states conclusions.

Author reply:

Your valuable comments are very conducive to the reader's understanding of this paper. We have minimized the use of acronyms and given the detailed explanations on the physical meaning of electron localization function (ELF) and crystal orbital Hamilton population (COHP) in our revised manuscript. At the same time, we also made an in-depth discussion about how they support our claims. Electron localization function (ELF) is a measure of relative electron localization,^{6,7} and it maps values in the range from 0 to 1, where 0.5 represents the situation in an homogeneous electron gas. The large ELF values usually occur in regions with a high tendency of formation of electron pairs, corresponding to bonds, lone pairs electrons and electron shells.^{8,9} As shown in Fig. S6a, the maximum ELF values of ~ 0.9 between the iodine and nitrogen atoms indicates the existence of covalent bonds. And crystal orbital Hamilton population (COHP) is an energy-resolved partitioning scheme of the band structure energy on the basis of atomic and bonding contributions, where negative values indicate bonding, and positive values indicate antibonding behavior.^{10,11} Considering all valence orbitals, the bond strength between two interacting atoms can be visualized by investigating the complete COHP between them, and the integrated COHPs (ICOHPs) is used as a qualitative measure of mainly covalent bond strength, where the greater negative value, the stronger the covalent interaction.¹² As shown in Fig. 3b, the negative COHPs observed below the Fermi level mean that N–N and I–N interactions are responsible for structural stability. And the resulting ICOHPs of N–N, and I–N pairs up to the Fermi level are -12.75 , and -2.04 eV, respectively, which further confirms that strong covalent bonding interactions occur between I and N atoms. We have redone the figures according to the revised content, as shown Fig. 3

and Fig. 4.

Fig. 3. **a** Projected density of states (PDOS) in $R\bar{3}m$ - IN_6 at 100 GPa. The Fermi energy is set to zero. **b** COHP of N–N, I–N, and I–I pairs in $R\bar{3}m$ - IN_6 . **c** The projected COHP of I–N pairs. Positive and negative $-p$ COHP values denote bonding and antibonding interactions, respectively. **d** the MO energy level diagram of the icosahedral cage-like IN_{12} molecular fragment with D_{3d} point group symmetry. Besides grey annotated MOs of icosahedral cage-like IN_{12} , others mainly consist of I and N_6 ring components.

Fig. 4. **a** $-ICOHP$ s of the M–N pairs in $R\bar{3}m$ - MN_6 ($M = F, Cl, Br, \text{ and } I$) compounds at 100 GPa. The optimized atomic geometries are shown in the insets. **b** Pressure-induced variations in $-ICOHP$ and bond lengths for the two types of I–N bonds in $R\bar{3}m$ - IN_6 . A positive $-ICOHP$ indicates the presence of bonding states.

Responses to reviewer #3

Comments to the Author

This manuscript presents DFT-predictions of iodine-nitrogen compounds stabilized under high pressure conditions. The structure mostly focused on can be formally thought of as 2^- rings that are coordinated by 2^+ cations. I have reviewed this work once before, for [redacted], and the authors may recognize some of my comments. Whereas the predicted chemical structure is quite interesting, the analysis of the electronic structure underlying it is subpar. The authors attempt to throw everything at the poor compound: MO theory, ELF, VSEPR, ideas of octet rules and electronegativity, radii, analyses of the DOS, orbital hybridization, and QTAIM. Unfortunately, a rather shallow understanding of all of the mentioned theories and methods are demonstrated. It would strengthen the work if the authors better learned and focused on a few perspectives of analyzing electronic structure rather than be all over the map. The text is full of statements and loose conclusions that are not supported by calculations or proper explanations. As a consequence, I cannot recommend publication of this work.

Author reply:

We are very grateful to you for reviewing our manuscript again. Our manuscript was previously submitted to [redacted] and has been revised according to your comments. We are very sorry that the revision can't address all your comments very well. We highly agree with your valuable comments "it would strengthen the work if the authors better learned and focused on a few perspectives of analyzing electronic structure rather than be all over the map." This time, according to your suggestions, we rethought our perspectives and chosen the molecular orbitals (MOs) theory to conduct in-depth analysis for the electronic structure and made substantial revisions for the manuscript. We hope the revised manuscript would address your comments.

1. It is claimed that iodine behaves as a transition metal under pressure. Yes, it would appear that I $5d$ levels may become occupied. Based on previous work on the topic of compressed heavy main group elements, the latter is expected. However, the bonding of I is still predominately governed by p -states, which is not the case for any transition metal. The DOS is (as expected) dominated by the $2p$ levels of nitrogen. And whereas the $5d$ come down with pressure, and the interaction with N $2p$ levels is now quantified to some degree, why do these interactions give rise to 12 fold-coordination? Is the $5d$ really essential for this?

Author reply:

Iodine, as a member of halogen elements, has a valence configuration of $5s^25p^5$. In general, the $5s$ electrons are regarded as the inner valence electrons, and $5p$ electrons are extensively used in bonding. The p states are very important for the bonding interaction. Therefore, p states have larger contribution to the DOS. However, as shown in the projected density of states (PDOS) of $R\bar{3}m$ - IN_6 , there is the contribution of I $5d$ below the Fermi level, indicating that it becomes a valence electron orbital and participates in bonding between nitrogen and iodine atoms. The detailed COHP calculations also confirm that there are bonding interactions between I $5d$ and nitrogen (Fig. 3). Previous works on the topic of compressed heavy main group elements (such as K and Cs) have also shown that the d levels could be occupied.¹³⁻¹⁵ The vacant d orbitals participating in bonding interaction is a part feature of the transition metal. However, the participation of $5d$ in the $R\bar{3}m$ - IN_6 is caused by the high pressure and the presence of N_6 rings, and its contribution is not as large as the intrinsic p states. As you said, this is slightly different from the transition metal. Therefore, we have corrected the claim in our revised manuscript.

The I $5d$ orbital is essential for the formation of the 12-fold coordination. To answer this question, we reorganized the manuscript according to your valuable suggestions (Questions 4). The molecular orbitals (MOs) are employed to explain the effect of I $5d$ orbital on the formation of the 12-fold coordination (Fig. 3d).

Fig. 2. **a** Crystal structure of $R\bar{3}m$ - IN_6 at 100 GPa. **b** The twelve-fold coordination of iodine (IN_{12}) coordinated with nitrogen atoms from eight armchair-like N_6 rings. **c** Schematic representation of the IN_{12} icosahedron structure with D_{3d} point group symmetry. N1 and N2 represent the nitrogen atoms in the short and long I–N bonds, respectively.

Fig. 3. **a** Projected density of states (PDOS) in $R\bar{3}m$ - IN_6 at 100 GPa. The Fermi energy is set to zero. **b** COHP of N–N, I–N, and I–I pairs in $R\bar{3}m$ - IN_6 . **c** The projected COHP of I–N pairs. Positive and negative $-p$ COHP values denote bonding and antibonding interactions, respectively. **d** the MO energy level diagram of the icosahedral cage-like IN_{12} molecular fragment with D_{3d} point group symmetry. Besides grey annotated MOs of icosahedral cage-like IN_{12} , others mainly consist of I and N_6 ring components.

To gain insight into the formation mechanism of hypercoordinated IN_6 under high pressure, we investigated the composition of molecular orbitals (MOs) for I $5s$, $5p$ and $5d$ orbitals interacting with N $2s$ and $2p$ orbitals in an icosahedral cage-like IN_{12} fragment. We cut the IN_{12} molecular fragment from the crystal structure of $R\bar{3}m$ - IN_6 . The symmetry analysis indicates that the IN_{12} molecular fragment, consisted of six N1 and six N2 atoms, has a D_{3d} point group symmetry (Fig. 2c). Therefore, according to the point group symmetry D_{3d} , the MOs of the icosahedral cage-like IN_{12} fragment can be described by the orbital interaction of I and N_6 ring (see Fig. S5). The valence electrons of iodine and nitrogen participate in the formation of chemical bonds in $R\bar{3}m$ - IN_6 . Two valence electrons of nitrogen form two N–N single bonds $\sigma(sp^3, sp^3)$ with two neighbouring nitrogen atoms in the N_6 ring. The remaining three electrons of the nitrogen atom in two sp^3 orbitals are used to interact with the iodine atom. Therefore, a total of twenty-five valence electrons occupies the MOs of the icosahedral cage-like IN_{12} fragment in the $R\bar{3}m$ - IN_6 according to the rules of the Aufbau principle (filling from lowest to highest energy), Hund’s rules (maximum spin multiplicity consistent with the lowest net energy), and the Pauli exclusion principle (no two electrons with identical quantum numbers). The component analysis of electron occupied MOs is employed by the Amsterdam density functional (ADF)

package¹⁶, and MO energy level diagram is shown in Fig. 3d (the detail also see Table S3). It is noteworthy that the designations of the bonding, nonbonding, and antibonding nature are based on the symmetry of orbitals, which will not change under high pressure. The MOs energy level diagram of the icosahedral cage-like IN₁₂ fragment reveals that the valence electrons of I and N in N₆ ring have a strong participation in the MOs of the IN₁₂ fragment containing 25 electrons. Note that nine MOs of IN₁₂ fragment at the high-lying energy level mainly originate from the combination of the E_g and A_{1g} orbitals of I 5*d* and N₆ ring component. Similarly, the E_u and A_{2u} orbitals of I 5*p* also participate in the orbital interaction with the orbitals of N₆ ring component. The A_{1g} orbitals of I 5*s* and N₆ ring component form bonding A_{1g} and antibonding A_{1g}^{*} orbitals of IN₁₂ fragment at the low-lying energy levels. These results confirm that I 5*d* orbitals become a part of the valence orbitals and participate in the bonding interactions. The unique highly symmetrical icosahedral N₁₂ cage and the participation of I 5*d* orbitals in the bonding interactions contribute to the formation of twelve-fold iodine hypercoordination. Therefore, the I 5*d* orbitals are essential for 12 fold-coordination.

Table S3 The component analysis of electron occupied molecular orbitals (MOs) of IN₁₂ in Figure 3d of main text. Due to D_{3d} point group symmetry of the structure, twelve N atoms have two positions and they are divided into two compositions. The composition of six N atoms, whose position is closer to the I atom, is called 6N1 (six N1 atoms). The other composition is called 6N2 (six N2 atoms). The E represents the energy of MOs and the unit is eV. The OS and os represent the orbital symmetry of IN₁₂ and compositions (I, 6N1, 6N2), respectively. The c is composition of IN₁₂.

Spin up						Spin down					
MOs	E (eV)	OS	percentage	os	c	MOs	E (eV)	OS	percentage	os	c
HOMO	Not selected					HOMO-1	-7.30	A _{1.g}	38.57%	A _{1.g}	6N1
									37.82%	A _{1.g}	6N1
									22.64%	A _{1.g}	6N2
									1.57%	5 d	I
									1.11%	5 s	I
HOMO-2	-7.49	E _{1.g:2}	39.45%	E _{1.g:2}	6N1	HOMO-2	-7.39	E _{1.g:2}	45.15%	E _{1.g:2}	6N2
			24.67%	E _{1.g:2}	6N2				26.76%	E _{1.g:2}	6N1
			16.50%	E _{1.g:2}	6N2				12.99%	E _{1.g:2}	6N2
			13.58%	E _{1.g:2}	6N1				6.24%	E _{1.g:2}	6N1
			3.39%	E _{1.g:2}	6N1				5.34%	E _{1.g:2}	6N2
			1.56%	E _{1.g:2}	6N2				3.95%	E _{1.g:2}	6N1
			1.07%	5 d	I						

	HOMO-3	-7.49	E1.g:1	39.45%	E1.g:1	6N1		HOMO-3	-7.39	E1.g:1	45.15%	E1.g:1	6N2
	24.67%	E1.g:1	6N2	26.76%	E1.g:1	6N1							
	16.50%	E1.g:1	6N2	12.99%	E1.g:1	6N2							
	13.58%	E1.g:1	6N1	6.24%	E1.g:1	6N1							
	3.39%	E1.g:1	6N1	5.34%	E1.g:1	6N2							
	1.56%	E1.g:1	6N2	3.95%	E1.g:1	6N1							
	1.07%	5d	I										
	HOMO-4	-7.89	E1.g:2	46.57%	E1.g:2	6N1		HOMO-8	-9.79	E1.g:2	46.27%	E1.g:2	6N1
	22.82%	E1.g:2	6N2	42.40%	E1.g:2	6N2							
	13.23%	E1.g:2	6N2	4.51%	E1.g:2	6N2							
	9.98%	E1.g:2	6N2	1.77%	E1.g:2	6N1							
	4.01%	E1.g:2	6N1	1.72%	E1.g:2	6N2							
	3.06%	E1.g:2	6N1	1.05%	5d	I							
	HOMO-5	-7.89	E1.g:1	46.57%	E1.g:1	6N1		HOMO-9	-9.79	E1.g:1	46.27%	E1.g:1	6N1
	22.82%	E1.g:1	6N2	42.40%	E1.g:1	6N2							
	13.23%	E1.g:1	6N2	4.51%	E1.g:1	6N2							
	9.98%	E1.g:1	6N2	1.77%	E1.g:1	6N1							
	4.01%	E1.g:1	6N1	1.72%	E1.g:1	6N2							
	3.06%	E1.g:1	6N1	1.05%	5d	I							
	HOMO-9	-9.53	A1.u	86.14%	A1.u	6N1		HOMO-11	-10.17	E1.g:2	79.31%	E1.g:2	6N1
	13.49%	A1.u	6N2	10.13%	E1.g:2	6N2							
				2.88%	E1.g:2	6N2							
				2.06%	E1.g:2	6N1							
	HOMO-10	-9.84	E1.g:2	34.01%	E1.g:2	6N1		HOMO-12	-10.17	E1.g:1	79.31%	E1.g:1	6N1
	32.89%	E1.g:2	6N2	10.13%	E1.g:1	6N2							
	21.12%	E1.g:2	6N2	2.88%	E1.g:1	6N2							
	5.63%	E1.g:2	6N1	2.06%	E1.g:1	6N1							
	1.91%	E1.g:2	6N1	1.85%	E1.g:1	6N2							
	1.33%	5d	I	1.26%	5d	I							
	1.30%	E1.g:2	6N2										
	HOMO-11	-9.84	E1.g:1	34.01%	E1.g:1	6N1		HOMO-15	-10.30	A1.u	94.38%	A1.u	6N1
	32.89%	E1.g:1	6N2	5.40%	A1.u	6N2							
	21.12%	E1.g:1	6N2										
	5.63%	E1.g:1	6N1										
	1.91%	E1.g:1	6N1										
	1.33%	5d	I										
1.30%	E1.g:1	6N2											
	HOMO-18	-12.46	E1.u:2	28.39%	5p	I		HOMO-17	-12.50	A2.u	33.81%	5p	I
	22.09%	E1.u:2	6N1	26.53%	A2.u	6N2							
	16.01%	E1.u:2	6N1	16.66%	A2.u	6N2							
	12.86%	E1.u:2	6N2	10.55%	A2.u	6N2							
	7.49%	6p	I	7.94%	6p	I							

			6.84%	E1.u:2	6N1				3.07%	A2.u	6N ₁
			3.98%	E1.u:2	6N2				1.73%	A2.u	6N1
			1.99%	E1.u:2	6N2				1.21%	A2.u	6N1
			1.74%	E1.u:2	6N2						
HOMO-19	-12.46	E1.u:1	28.39%	5p	I	HOMO-18	-12.54	E1.u:2	34.81%	E1.u:2	6N1
			22.09%	E1.u:1	6N1				30.57%	5p	I
			16.01%	E1.u:1	6N1				11.37%	E1.u:2	6N1
			12.86%	E1.u:1	6N2				7.02%	6p	I
			7.49%	6p	I				6.05%	E1.u:2	6N2
			6.84%	E1.u:1	6N1				5.62%	E1.u:2	6N2
			3.98%	E1.u:1	6N2				4.04%	E1.u:2	6N1
			1.99%	E1.u:1	6N2						
			1.74%	E1.u:1	6N2						
HOMO-20	-13.48	A2.u	34.40%	A2.u	6N2	HOMO-19	-12.54	E1.u:1	34.81%	E1.u:1	6N1
			27.56%	5p	I				30.57%	5p	I
			15.34%	A2.u	6N2				11.37%	E1.u:1	6N1
			12.86%	A2.u	6N2				7.02%	6p	I
			6.26%	6p	I				6.05%	E1.u:1	6N2
			2.37%	A2.u	6N1				5.62%	E1.u:1	6N2
			1.56%	A2.u	6N1				4.04%	E1.u:1	6N1
			1.01%	A2.u	6N1						
HOMO-21	-17.73	A1.g	55.86%	A1.g	6N1	HOMO-22	-18.08	A1.g	31.36%	5s	I
			24.88%	5s	I				30.88%	A1.g	6N1
			6.25%	A1.g	6N2				24.25%	A1.g	6N2
			5.40%	6s	I				5.84%	6s	I
			3.27%	A1.g	6N2				4.75%	A1.g	6N1
			3.14%	A1.g	6N1				1.61%	A1.g	6N2
			1.63%	A1.g	6N2				1.38%	A1.g	6N2
HOMO-33	-26.58	A1.g	46.72%	5s	I	HOMO-32	-26.41	A1.g	49.60%	5s	I
			30.49%	A1.g	6N2				24.78%	A1.g	6N1
			17.25%	A1.g	6N1				20.77%	A1.g	6N2
			4.62%	A2.g	6N2				4.14%	A2.g	6N1
			3.69%	A1.g	6N1				3.64%	A1.g	6N2

Here, the molecular orbitals in list are characterized by a composition of more than 30% of 6N1 composition or I atoms and a large contribution of the p orbitals of 6N1. Although there are the other MOs with a large proportion of 6N1, they are made up more of s orbital or vacant orbital of N atoms than p orbitals.

2. It is in this work (still) claimed that neutral N₆ is planar, but that is not true. There is extensive literature on this topic (N₆ chemistry). Higher levels of theory show that the planar geometry is unstable. Why do the authors think the neutral molecule is

planar? Could it be they simply compute it in that geometry at a low level of theory, and don't check the alternatives, or read the literature?

Author reply

We are very sorry for the wrong expression of N_6 rings. There are relevant literatures on the this topic, Ha *et al.* reported “the free N_6 is not stable”.¹⁷ And Huber *et al.* also pointed out “For as long as no direct experimental proof for the existence of N_6 exists we must assume that hexazine is neither thermodynamically nor kinetically stable”.¹⁸ In this paper, we want to express the hypothetical planar hexazine N_6 ring as expressed in *Phys Rev Lett* **126**, 065702 (2021).¹⁹ The expressions involved in the manuscript have been fully corrected.

3. The authors classify the N-I bonding as:“a coordinate covalent (dative bond) is formed, which is a kind of two-center-two electron covalent bond in which the two bonding electrons are contributed by a single atom”. Granted this used be even more confused in the last version of the manuscript, I still do not follow. On page 5 it is claimed that three of the $5d$ orbitals of I are vacant and can serve as electron acceptors for the lone pair electrons on the N_6 rings. At the same time, the authors show how the I atom is oxidized (but that is mentioned elsewhere). And why only three orbitals? All the $5d$ are formally unoccupied in the isolated atom, but in the extended system they all form a band, and some degree of all of them become occupied in the interaction with N. Iodine is further claimed to exhibit sp^3d^2 hybridization, but this is not substantiated in any way. The discussion regarding the “octahedral split” is confusing at best. Which are the directions indicated, and how do they relate to the coordination geometry?

Author reply:

Thank you for your kind suggestions. We have removed the related claims about the coordinate covalent bond, sp^3d^2 hybridization and the octahedral split, and choose to conduct in-depth analysis from the perspective of molecular orbitals (MOs). Although great revision has been made, it does not affect the overall framework of the manuscript, and the main purpose is still to explain the formation mechanism of hypercoordinated IN_6 . As you mentioned, the I $5d$ are formally unoccupied in the isolated atom, but in the I–N system they are occupied in the interaction with N_{12} cage. This can be explained by the decomposition of COHP into atomic orbitals, in which the negative COHP values below the Fermi level show that all of the five d orbitals contribute to the bonding.

Fig. R1. Projected density of states (PDOS) and crystal orbital Hamilton population (COHP) in $R\bar{3}m$ - IN_6 at 100 GPa. Positive and negative $-pCOHP$ values denote bonding and antibonding interactions, respectively.

Fig. 3. **a** Projected density of states (PDOS) in $R\bar{3}m$ - IN_6 at 100 GPa. The Fermi energy is set to zero. **b** COHP of N-N, I-N, and I-I pairs in $R\bar{3}m$ - IN_6 . **c** The projected COHP of I-N pairs. Positive and negative $-pCOHP$ values denote bonding and antibonding interactions, respectively. **d** the MO energy level diagram of the icosahedral cage-like IN_{12} molecular fragment with D_{3d} point group symmetry. Besides grey annotated MOs of icosahedral cage-like IN_{12} , others mainly consist of I and N_6 ring components.

Furthermore, in this time we choose to employ the MOs to explain the occupation of I

5*d* orbitals (please refer to the reply for Questions 1). We investigated the composition of molecular orbitals (MOs) for I 5*s*, 5*p* and 5*d* orbitals interacting with N 2*s* and 2*p* orbitals in an icosahedral cage-like IN₁₂ fragment. We cut the IN₁₂ molecular fragment from the crystal structure of *R* $\bar{3}m$ -IN₆. The symmetry analysis indicates that the IN₁₂ molecular fragment, consisted of six N1 and six N2 atoms, has a D_{3d} point group symmetry (Fig. 2c). Therefore, according to the point group symmetry D_{3d}, the MOs of the icosahedral cage-like IN₁₂ fragment can be described by the orbital interaction of I and N₆ ring (see Fig. S5). The valence electrons of iodine and nitrogen participate in the formation of chemical bonds in *R* $\bar{3}m$ -IN₆. Two valence electrons of nitrogen form two N–N single bonds $\sigma(sp^3, sp^3)$ with two neighbouring nitrogen atoms in the N₆ ring. The remaining three electrons of the nitrogen atom in two *sp*³ orbitals are used to interact with the iodine atom. Therefore, a total of twenty-five valence electrons occupies the MOs of the icosahedral cage-like IN₁₂ fragment in the *R* $\bar{3}m$ -IN₆ according to the rules of the Aufbau principle (filling from lowest to highest energy), Hund's rules (maximum spin multiplicity consistent with the lowest net energy), and the Pauli exclusion principle (no two electrons with identical quantum numbers). The component analysis of electron occupied MOs is employed by the Amsterdam density functional (ADF) package¹⁶, and MO energy level diagram is shown in Fig. 3d (the detail also see Table S3). It is noteworthy that the designations of the bonding, nonbonding, and antibonding nature are based on the symmetry of orbitals, which will not change under high pressure. The MOs energy level diagram of the icosahedral cage-like IN₁₂ fragment reveals that the valence electrons of I and N in N₆ ring have a strong participation in the MOs of the IN₁₂ fragment containing 25 electrons. Note that nine MOs of IN₁₂ fragment at the high-lying energy level mainly originate from the combination of the E_g and A_{1g} orbitals of I 5*d* and N₆ ring component. Similarly, the E_u and A_{2u} orbitals of I 5*p* also participate in the orbital interaction with the orbitals of N₆ ring component. The A_{1g} orbitals of I 5*s* and N₆ ring component form bonding A_{1g} and antibonding A_{1g}^{*} orbitals of IN₁₂ fragment at the low-lying energy levels. These results confirm that I 5*d* orbitals become a part of the valence orbitals and participate in the bonding interactions. The unique highly symmetrical icosahedral N₁₂ cage and the participation of I 5*d* orbitals in the bonding interactions contribute to the formation of twelve-fold iodine hypercoordination. The I 5*d* orbitals are essential for 12 fold-coordination.

4. Figure 3 shows some sort of orbital splitting diagram (where phases have been omitted). But this diagram is not connected in any relevant way to the analysis, nor is it apparent how it could be. The SI shows an MO diagram of the π -system of the (unstable) planar N₆, and how the addition of 2e- and the puckering, confusingly enough, does absolutely nothing to the MOs. I recommend the authors instead provide

a figure of the puckered N_6 and attempt to explain the effect of occupation by 2 extra electrons. If the authors are able, it would strengthen the work if the interactions of these levels with the I $5p$ and $5d$ shell could be shown explicitly in a model compound. If they are not able, which I suspect, then I suggest they instead forgo trying to discuss the bonding in terms of interacting orbitals. Maybe instead more matter of fact list atomic charges and focus on describing the geometry. The COHP looks in order, but the conclusions drawn from it are not, or are taken too far. Simply stating that there is $5d$ orbital occupation may be enough, and that there is bonding interactions between N and I. The work would be cleaner, and less filled with unsubstantiated claims, if it simply communicated: “we predict that this compound, special in many ways, should be stable at high pressure. Here are some basic bonding properties”. That would be interesting to know for experimentalists. The work could then be shortened by half. One way or another, to be acceptable for publication, this work needs to be significantly toned down and clarified.

Author reply:

According to your kind suggestions, we have made substantial revisions to the manuscript, and have removed the claims and diagrams related to octahedral split and the MOs of the π -system of the N_6 ring. We reorganized the manuscript according to your valuable suggestions. The molecular orbitals (MOs) are employed to explain the effect of I $5d$ orbital on the formation of the 12 fold-coordination (Fig. 3d).

Fig. 3. **a** Projected density of states (PDOS) in $R\bar{3}m$ - IN_6 at 100 GPa. The Fermi energy is set to zero. **b** COHP of N–N, I–N, and I–I pairs in $R\bar{3}m$ - IN_6 . **c** The projected COHP of I–N pairs. Positive and negative $-p$ COHP values denote bonding and antibonding interactions, respectively. **d** the MO energy level diagram of the icosahedral cage-like IN_{12} molecular fragment with D_{3d} point group symmetry.

Besides grey annotated MOs of icosahedral cage-like IN_{12} , others mainly consist of I and N_6 ring components.

We investigated the composition of molecular orbitals (MOs) for I $5s$, $5p$ and $5d$ orbitals interacting with N $2s$ and $2p$ orbitals in an icosahedral cage-like IN_{12} fragment. We cut the IN_{12} molecular fragment from the crystal structure of $R\bar{3}m\text{-IN}_6$. The symmetry analysis indicates that the IN_{12} molecular fragment, consisted of six N1 and six N2 atoms, has a D_{3d} point group symmetry (Fig. 2c). Therefore, according to the point group symmetry D_{3d} , the MOs of the icosahedral cage-like IN_{12} fragment can be described by the orbital interaction of I and N_6 ring (see Fig. S5). The valence electrons of iodine and nitrogen participate in the formation of chemical bonds in $R\bar{3}m\text{-IN}_6$. Two valence electrons of nitrogen form two N–N single bonds $\sigma(sp^3, sp^3)$ with two neighbouring nitrogen atoms in the N_6 ring. The remaining three electrons of the nitrogen atom in two sp^3 orbitals are used to interact with the iodine atom. Therefore, a total of twenty-five valence electrons occupies the MOs of the icosahedral cage-like IN_{12} fragment in the $R\bar{3}m\text{-IN}_6$ according to the rules of the Aufbau principle (filling from lowest to highest energy), Hund's rules (maximum spin multiplicity consistent with the lowest net energy), and the Pauli exclusion principle (no two electrons with identical quantum numbers). The component analysis of electron occupied MOs is employed by the Amsterdam density functional (ADF) package¹⁶, and MO energy level diagram is shown in Fig. 3d (the detail also see Table S3). It is noteworthy that the designations of the bonding, nonbonding, and antibonding nature are based on the symmetry of orbitals, which will not change under high pressure. The MOs energy level diagram of the icosahedral cage-like IN_{12} fragment reveals that the valence electrons of I and N in N_6 ring have a strong participation in the MOs of the IN_{12} fragment containing 25 electrons. Note that nine MOs of IN_{12} fragment at the high-lying energy level mainly originate from the combination of the E_g and A_{1g} orbitals of I $5d$ and N_6 ring component. Similarly, the E_u and A_{2u} orbitals of I $5p$ also participate in the orbital interaction with the orbitals of N_6 ring component. The A_{1g} orbitals of I $5s$ and N_6 ring component form bonding A_{1g} and antibonding A_{1g}^* orbitals of IN_{12} fragment at the low-lying energy levels. These results confirm that I $5d$ orbitals become a part of the valence orbitals and participate in the bonding interactions. The unique highly symmetrical icosahedral N_{12} cage and the participation of I $5d$ orbitals in the bonding interactions contribute to the formation of twelve-fold iodine hypercoordination. The I $5d$ orbitals are essential for 12 fold-coordination.

5. The literature is lacking overall. To name just a few examples: Where does the covalent radii come from? Is there any work that has validated the use of the Laplacian of the electron density as an indicator for covalent bonds under high pressure? One could at minimum expect a citation to work showing this under ambient conditions. The lacking N₆ references.

Author reply:

We have added some reference materials in our revised manuscript. The covalent radii used in the manuscript is from *Dalton Trans.* 2008, (21), 2832-2838.²⁰ Previous works have shown that the Laplacian of the electron density $\nabla^2\rho(\mathbf{r})$ values at critical points can efficiently reflect the strength of covalent bonds, Li *et al.* used the $\nabla^2\rho(\mathbf{r})$ values to show the covalent interactions between nitrogen atoms in HeN₄ at ambient pressure.²¹ The negative $\nabla^2\rho(\mathbf{r})$ values calculated by Peng *et al.* proves that the formation of N–N and N–Xe covalent bonds in XeN₆ at 150 GPa.²² Liu *et al.* reported obvious H–Se covalent bonds in the predicted H₆SSe at 200 GPa by using the Laplacian of the electron density.²³ In addition, Liu *et al.* concluded from topological properties that the He···O interaction at 300 GPa is a strong, closed-shell, bonding interaction with the character and strength similar to that of hydrogen bonds by using the Laplacian of the electron density.²⁴ Furthermore, we have made an overview about the polymeric N₆ ring, Peng *et al.* predicted XeN₆ with the appearance of chaired N₆ hexagons in 2015.²² Subsequently, the WN₆ with armchair like N₆ rings have been predicted at 65 GPa²⁵ and then been successfully synthesized.¹⁹ Recently, TeN₆ with armchair-like cyclo-N₆ anions has been described.²⁶ All of these references have been added to the revised manuscript.

6. It is claimed that the described compound is a potential energetic material. What happens to the discussed structure when it is relaxed to ambient conditions? Is it still kinetically stable? If not, all such claims should be removed. And if it is stable at ambient conditions, how is the energy density computed? If you compare energy densities of a compressed material with conventional explosives at ambient conditions, it is hardly a fair comparison.

Author reply:

The predicted IN₆ emerges at the high pressure of 100 GPa and is enthalpically favorable over a pressure range of 100–150 GPa. In order to discuss the dynamical stability of the IN₆, we perform the phonon calculations in the pressure range of 0–200 GPa, and the imaginary frequency does not disappear above 40 GPa, indicating

that the $R\bar{3}m$ -IN₆ structure is only dynamically stable at >40 GPa (Fig. S3). Although $R\bar{3}m$ -IN₆ can not be recovered to ambient pressure, it is expected to decompose exothermically to solid I₂ and molecular N₂ and still has the potential to be high-energy-density material. Similar claims have been reported before, XeN₆ was only dynamically stable at >50 GPa and estimated to have an energy density of approximately 2.4 kJ g⁻¹.²² And Zn(N₅)₂ had dynamic stability at 25–85 GPa with an energy density of 6.57 kJ g⁻¹.²⁷ For the IN₆, its released energy density is approximately 6.91 kJ·g⁻¹. And we agree with your suggestions. It is unfair to compare energy densities of compressed material with that of conventional explosives at ambient conditions. And in this paper, we mainly focus on the bonding interaction between iodine and nitrogen. Therefore, we have removed the relevant claims in our revised manuscript.

Figure. S3 Phonon dispersion curves of $R\bar{3}m$ -IN₆.

7. What are the I and N reference phases used for the construction of the convex hull? I hope they are not the same throughout, since the ground state of both elements change with compression. This needs to be specified.

Author reply:

We are very sorry that we neglected to provide the reference phases of I and N used for the construction of the convex hull. The corresponding stable phases of solid nitrogen and iodine as the reference structures, in which nitrogen adopted $Pa\bar{3}$, $P4_12_12$ and $I2_13$ structures,^{1,2} and the iodine adopted $Immm$, $I4/mmm$ and $Fm\bar{3}m$ structures.³⁻⁵ We have added them in our revised manuscript.

References

1. Eremets, M. I., Gavriluk, A. G., Trojan, I. A., Dzivenko, D. A. & Boehler, R. Single-bonded Cubic Form of Nitrogen. *Nat. Mater.* **3**, 558-563 (2004).
2. Pickard, C. J. & Needs, R. J. High-pressure Phases of Nitrogen. *Phys. Rev. Lett.* **102**, 125702 (2009).
3. Duan, D. *et al.* Effect of nonhydrostatic pressure on superconductivity of monatomic iodine: An *ab initio* study. *Phys. Rev. B* **79**, 064518 (2009).
4. Kume, T., Hiraoka, T., Ohya, Y., Sasaki, S. & Shimizu, H. High Pressure Raman Study of Bromine and Iodine: Soft Phonon in the Incommensurate Phase. *Phys. Rev. Lett.* **94**, 065506 (2005).
5. Kenichi, T., Kyoko, S., Hiroshi, F. & Mitsuko, O. Modulated structure of solid iodine during its molecular dissociation under high pressure. *Nature* **423**, 971-974 (2003).
6. Becke, A. D. & Edgecombe, K. E. A simple measure of electron localization in atomic and molecular systems. *J. Chem. Phys.* **92**, 5397-5403 (1990).
7. Lennard - Jones, J. E. The Spatial Correlation of Electrons in Molecules. *J. Chem. Phys.* **20**, 1024-1029 (1952).
8. Savin, A. *et al.* Electron Localization in Solid-State Structures of the Elements: the Diamond Structure. *Angew. Chem. Int. Ed. Engl.* **31**, 187-188 (1992).
9. Häussermann, U. *et al.* Localization of Electrons in Intermetallic Phases Containing Aluminum. *Angew. Chem., Int. Ed. Engl.* **33**, 2069-2073 (1994).
10. Dronskowski, R. & Blochl, P. E. Crystal Orbital Hamilton Populations (COHP). Energy-Resolved Visualization of Chemical Bonding in Solids Based on Density-Functional Calculations. *J. Phys. Chem.* **97**, 8617-8624 (1993).
11. Landrum, G. A. & Dronskowski, R. The Orbital Origins of Magnetism: From Atoms to Molecules to Ferromagnetic Alloys. *Angew. Chem. Int. Ed.* **39**, 1560-1585 (2000).
12. Maintz, S., Deringer, V. L., Tchougréeff, A. L. & Dronskowski, R. LOBSTER: A tool to extract chemical bonding from plane-wave based DFT. *J. Comput. Chem.* **37**, 1030-1035 (2016).
13. Parker, L. J., Atou, T. & Badding, J. V. Transition Element-Like Chemistry for Potassium Under Pressure. *Science* **273**, 95-97 (1996).
14. Delattre, J. L. & Badding, J. V. Rietveld Refinement of the Crystal Structure of Cs(IV), *ad*-Electron Metal. *J. Solid State Chem.* **144**, 16-19 (1999).
15. Dong, X. *et al.* How do chemical properties of the atoms change under pressure. *arXiv*, 1503.00230 (2015).
16. Balakrishnarajan, M. M., Pancharatna, P. D. & Hoffmann, R. Structure and bonding in

- boron carbide: The invincibility of imperfections. *New J. Chem.* **31**, 473-485 (2007).
17. Ha, T.-K., Cimiraglia, R. & Nguyen, M. T. Can hexazine (N₆) be stable? *Chem. Phys. Lett.* **83**, 317-319 (1981).
 18. Huber, H. Is Hexazine Stable? *Angew. Chem., Int. Ed. Engl.* **21**, 64-65 (1982).
 19. Salke, N. P. *et al.* Tungsten Hexanitride with Single-Bonded Armchairlike Hexazine Structure at High Pressure. *Phys. Rev. Lett.* **126**, 065702 (2021).
 20. Cordero, B. *et al.* Covalent radii revisited. *Dalton Trans.*, 2832-2838 (2008).
 21. Li, Y. *et al.* Route to high-energy density polymeric nitrogen *t*-N via He–N compounds. *Nat. Commun.* **9**, 722 (2018).
 22. Peng, F., Wang, Y., Wang, H., Zhang, Y. & Ma, Y. Stable xenon nitride at high pressures. *Phys. Rev. B* **92**, 094104 (2015).
 23. Liu, B. *et al.* Effect of covalent bonding on the superconducting critical temperature of the H-S-Se system. *Phys. Rev. B* **98**, 174101 (2018).
 24. Liu, H., Yao, Y. & Klug, D. D. Stable structures of He and H₂O at high pressure. *Phys. Rev. B* **91**, 014102 (2015).
 25. Xia, K. *et al.* A novel superhard tungsten nitride predicted by machine-learning accelerated crystal structure search. *Sci. Bull.* **63**, 817-824 (2018).
 26. Liu, Z. *et al.* Formation mechanism of insensitive tellurium hexanitride with armchair-like cyclo-N₆ anions. *Commun. Chem.* **3**, 42 (2020).
 27. Liu, Z. *et al.* Moderate Pressure Stabilized Pentazolate Cyclo-N₅[−] Anion in Zn(N₅)₂ Salt. *Inorg. Chem.* **59**, 8002-8012 (2020).

REVIEWERS' COMMENTS

Reviewer #2 (Remarks to the Author):

I feel the authors' revisions satisfy my criticisms and that the manuscript is now ready for publication in Nature Communications.

Reviewer #3 (Remarks to the Author):

It was a joy to read this revised version. The authors have carefully updated their work and removed many sticking points I previously objected to. Sometimes, less is more. I recommend publication and I look forward to the eventual experimental verification of these predictions.

Responses to Reviewer 2

Comments to the Author

I feel the authors' revisions satisfy my criticisms and that the manuscript is now ready for publication in Nature Communications.

Author reply:

We are very grateful for your review of our manuscript and very pleased with your recognition of our work.

Responses to Reviewer 3

Comments to the Author

It was a joy to read this revised version. The authors have carefully updated their work and removed many sticking points I previously objected to. Sometimes, less is more. I recommend publication and I look forward to the eventual experimental verification of these predictions.

Author reply:

Thank you very much for reviewing our manuscript again. Your constructive comments on our work in the previous reviews are crucial to improving the quality of this paper. We are very pleased that this revised manuscript is recognized.